# Personalized Nutrition in Chronic Kidney Disease

**DOI:** 10.3390/biomedicines13030647

**Published:** 2025-03-06

**Authors:** Nishigandha Pradhan, Jennifer Kerner, Luciana A. Campos, Mirela Dobre

**Affiliations:** 1School of Medicine, Case Western University, Cleveland, OH 44106, USA; 2Department of Medicine, Division of Nephrology and Hypertension, University Hospitals Cleveland Medical Center, Cleveland, OH 44106, USA; 3Center of Innovation, Technology and Education (CITE) at Anhembi Morumbi University—Anima Institute, São José dos Campos 12247-016, Brazil

**Keywords:** personalized nutrition, chronic kidney disease, renal diet

## Abstract

A personalized approach to nutrition in patients with chronic kidney disease (CKD) represents a promising paradigm shift in disease management, moving beyond traditional one-size-fits-all dietary recommendations. Patients with CKD often have other comorbidities and face unique nutritional challenges, including protein-energy wasting (PEW), sarcopenia, and impaired renal excretion of nutrients, which complicate dietary planning. Current guidelines focus primarily on nutrient restrictions—such as limiting protein, sodium, potassium, and phosphorus. However, these generalized recommendations often result in suboptimal adherence and outcomes. Personalized nutrition, which adapts dietary recommendations to individual characteristics, such as genotype, phenotype, and socio-cultural preferences, has gained traction across various chronic diseases. However, its application in nephrology remains underexplored, and despite promising results from studies such as Food4Me, questions remain about the real-world impact of such strategies. The aims of this review are (1) to summarize the evidence on the current state of nutritional recommendations in CKD, (2) to discuss the emerging role of multi-omics approaches in informing personalized nutrition advice in CKD, and (3) to provide an opinion on nutritional challenges faced by patients with CKD and the importance of collaboration with the renal dietician. We conclude that despite barriers, such as the cost and data integration, personalized nutrition holds the potential to improve CKD outcomes, enhance quality of life, and empower patients through tailored dietary strategies for better disease management.

## 1. Background

The global incidence and prevalence of chronic kidney disease (CKD) are increasing in parallel to rising prevalences of obesity, diabetes mellitus, hypertension, and metabolic syndrome [1]. All these conditions have many overlapping risk factors pertaining to diet and lifestyle, such as excessive caloric intake, consumption of ultraprocessed food, inadequate physical activity, and smoking. It is estimated that about 11 million deaths globally [2] and about half of all deaths due to cardiometabolic diseases in the US [3] can be attributed to poor diet. With an increase in the awareness of the central role played by diet in the development of these chronic diseases, there is greater focus on “food is medicine” [4,5]. The concept of personalized nutrition, which tailors dietary recommendations to individual characteristics, including genotype, phenotype, and socio-cultural factors, recognizes that individual responses to nutrition can vary significantly. Despite progress in understanding the complex interplay between diet and these individual factors in the course of some diseases, such as obesity [6,7], cardiovascular disease [8], metabolic syndrome [8], and diabetes [9,10], data in the field of kidney diseases remain limited. This review assesses the current state of nutritional guidelines for the management of CKD, the emerging role of multi-omics in informing personalized nutrition advice in CKD and highlights the associated nutritional challenges. It further explores the barriers to implementing personalized nutrition strategies in clinical practice for patients with CKD and outlines a potential framework for advancing individualized nutritional approaches to optimize CKD management on a patient-specific basis via two vignettes.

## 2. Emergence of Personalized Nutrition

Traditionally, nutritional advice has been targeted toward populations with some stratification based on broad characteristics like age, sex, and pregnancy status. The recognition that “one size does not fit all” and that individual responses to nutrition vary significantly due to intrinsic (such as genotype, epigenetic modifications, microbiome, and proteome) and extrinsic (including social determinants of health, behavioral patterns, and cultural and religious preferences) factors has led to the concept of “personalized nutrition”. Many other terms, such as precision nutrition, individualized nutrition, nutrigenetics, and nutrigenomics, are often used interchangeably, and a consensus on their definitions is lacking [11]. A simple definition proposed by Ordovas et al. defines personalized nutrition as “an approach that uses information on individual characteristics to develop targeted nutritional advice, products or services” [12].

The emergence of “omics” technologies—such as genomics, transcriptomics, proteomics, and metabolomics—along with advancements in “big data”, artificial intelligence, and systems biology, has fueled optimism about leveraging “precision nutrition” to provide highly personalized dietary recommendations for each individual [13].

However, data demonstrating the clinical benefit of such precise and high-cost dietary advice are limited. The largest randomized controlled trial attempting to address this question, the Food4Me Study, included 1269 adults from seven European countries [14]. Participants were randomized to the following four dietary advice groups: conventional, based on the individual baseline diet; based on the individual baseline diet plus phenotype; and based on the individual baseline diet plus phenotype plus genotype. While personalized nutrition was more effective than conventional dietary advice, incorporating phenotypic or genotypic information did not enhance the effectiveness of the personalized recommendations [14]. Examination of the impact of providing genotype-based personalized nutritional advice on individuals’ behavior found limited evidence supporting its effectiveness in motivating behavior change. Furthermore, genetic knowledge may have unintended negative consequences, such as increased demotivation and anxiety [15]. Similarly, a Cochrane meta-analysis reported no significant effect of communicating DNA-based risk estimates on dietary changes, highlighting the need for further investigation into the utility and psychological implications of such approaches [16].

Despite these limitations observed in the general population, we argue that personalized nutrition advice may hold promise in the management of complex patients with CKD, as the current nutritional guidelines for CKD fail to fully address their unique nutritional and medical challenges.

## 3. Current State of Nutritional Recommendations in CKD

With declining kidney function, the ability to excrete sodium and other electrolytes, water, acid, and other metabolic waste products is progressively impaired. Thus, traditional dietary recommendations for patients with CKD primarily focus on limiting the intake of potassium, sodium, and phosphorus to mitigate these challenges. Additionally, it is well-established that high protein intake contributes to glomerular hyperfiltration and elevated intraglomerular pressure, with further renal damage [17,18]. However, balancing all these restrictions, along with others that are needed for the management of common co-morbidities like diabetes mellitus, often leaves patients with limited options for a nutritionally adequate and satisfying diet. This frequently results in reduced adherence to dietary advice and greater frustration among patients. Emerging evidence suggests that shifting from rigid, nutrient-specific restrictions to promoting overall healthy dietary patterns, which help slow the progression of CKD, may be more feasible and acceptable for patients to follow [19,20].

### 3.1. Recommended Dietary Patterns in CKD Care

The Kidney Disease: Improving Global Outcomes (KDIGO) 2024 Clinical Practice Guideline for the Evaluation and Management of Chronic Kidney Disease recommends that patients with CKD adopt healthy and diverse diets with greater consumption of plant-based foods compared to animal-based foods and reduce the intake of ultraprocessed foods [21]. Additionally, recommendations emphasize the importance of nutrient bioavailability based on the food source, for example, recognizing that (added) inorganic phosphorus is absorbed more efficiently than organic (natural) dietary phosphorus [22]. This approach is supported by evidence suggesting that plant-based diets can reduce proteinuria, decrease metabolic acidosis, and slow CKD progression by reducing cardio-metabolic risk factors, such as hypertension, cardiovascular disease, diabetes, and obesity, without significantly impairing phosphorus control [21].

The American Diabetes Association (ADA) and KDIGO consensus report on diabetes management in CKD emphasizes the importance of individualized and balanced diets that are high in vegetables, fruits, and whole grains but low in refined carbohydrates and sugar-sweetened beverages. Both guidelines recommend a low-sodium diet (KDIGO < 2000 mg/day; ADA 1500 to <2300 mg/day) to control blood pressure and reduce cardiovascular risk [23]. Table 1 summarizes the types of diets recommended in CKD.

### 3.2. Energy Intake

The National Kidney Foundation Kidney Disease Outcomes Quality Initiative (KDOQI) recommends an energy intake of 25–35 kcal/kg body weight per day to maintain normal nutritional status in metabolically stable patients with CKD [22]. This recommendation is classified as Level 1, Grade C (low confidence) because of a lack of controlled metabolic studies and well-designed long-term clinical trials assessing energy intake among this population [22].

Energy intake should be individualized based on factors such as age, sex, activity level, body composition, weight management goals, CKD stage, inflammation, and comorbidities like diabetes mellitus, cardiovascular disease, and obesity. Additionally, consideration must be given to the risk of protein-energy wasting (PEW) and the need to control protein intake. In this context, carbohydrates generally make up about 50% of energy intake, with the rest from protein and fat [24].

### 3.3. Protein Intake

For dietary protein intake, the ADA and KDIGO recommend targeting 0.8 g/kg/day, while the KDOQI suggests 0.55–0.60 g/kg/day alone or 0.28–0.43 g/kg/day, with additional keto acid/amino acid analogues for metabolically stable patients with CKD without diabetes and 0.6–0.8 g/kg/day for those with diabetes and CKD [23,25]. Higher protein intakes should be considered for patients on maintenance dialysis because of the risk of malnutrition [23]. This recommendation is based on the observation from animal studies that a lower protein intake causes afferent arteriolar vasoconstriction, reducing the intraglomerular pressure and hyperfiltration injury [26]. Although the results of human studies on protein-restricted diets have been mixed, the overall evidence supports the role of reduced protein intake in limiting glomerular injury and proteinuria [27]. Reducing protein intake, particularly from animal sources, has been shown to lower the production of uremic toxins. This can help delay the need for renal replacement therapy and alleviate uremic symptoms in patients opting for conservative management [28].

Although clinical practice guidelines do not specify the optimal protein source (plant- or animal-based) citing insufficient evidence from randomized controlled trials, observational studies increasingly support the effectiveness of plant-based proteins in reducing the incidence and progression of CKD [29,30]. This has led the International Society of Renal Nutrition and Metabolism to recommend that “…it may be reasonable for nutrition professionals to suggest plant-based proteins with consideration of patient preferences and cultural food habits” [31].

This aligns with the observation that many “healthy dietary patterns” associated with a lower risk of incident CKD [32] and slower progression of established CKD [19] consist of a diet rich in fruits, vegetables, and nuts and are lower in red meats and processed foods.

### 3.4. Acid Load, Phosphorus, and Potassium Balance

Higher net endogenous acid production (NEAP) is associated with faster CKD progression. Current clinical guidelines recommend reducing the NEAP by means of greater dietary intake of fruits and vegetables or by oral alkali therapy [22]. Given the limited and mixed evidence regarding the effect of hyperphosphatemia on CKD progression, current guidelines only recommend “adjusting dietary phosphorus intake to maintain serum phosphorus in the normal range” and to consider differing bioavailability from various dietary sources [22]. Similarly, there was a weak recommendation (opinion) to adjust dietary potassium intake to maintain serum potassium within the normal range. Thus, under the supervision of a skilled renal dietician, it is possible to implement a low-protein, plant-based diet with a low risk of inducing muscle wasting, frailty, hyperphosphatemia, and hyperkalemia.

### 3.5. Omega-3 Polyunsaturated Fatty Acids (n-3 PUFAs) and Kidney Disease

Studies in animal models of chronic renal failure showed reduced oxidative stress, inflammation, and fibrosis leading to reduced tubulointerstitial injury after supplementation with docosahexaenoic acid (DHA) and eicosapentaenoic acid (EPA) [33]. In a recent pooled analysis of 25,570 participants from 19 cohorts in 12 countries measuring n-3 PUFA biomarker data, higher levels of total seafood n-3 PUFAs but not plant n-3 PUFAs were associated with a lower risk of incident CKD and slower decline in kidney function [34]. Systematic reviews of randomized controlled trials of PUFA-3 supplementation found a signal of an improved lipid profile and reduced oxidative stress, as well as a possible beneficial effect on preventing ESKD in patients with CKD not yet on renal replacement therapy, albeit with very low evidence certainty [35]. In the absence of more definitive results from large, randomized, controlled clinical trials, current clinical guidelines do not recommend the use of n-3 PUFA supplementation to prevent further decline in renal function in CKD though they do recommend the inclusion of 250 mg/day of long chain n-3 PUFAs from oily fish for the general population for a healthy diet [36].

## 4. Unique Nutritional Challenges in CKD

Patients with CKD face unique challenges, which make the personalization of nutrition recommendations essential (Figure 1). These challenges include protein-energy wasting (PEW), sarcopenia, the “Obesity paradox”, and specific considerations for older adults with CKD. Addressing these complexities requires a nuanced approach that considers individual patient characteristics, comorbidities, and lifestyle factors.

### 4.1. Protein-Energy Wasting and Sarcopenia in CKD

Declining kidney function is often associated with protein-energy wasting (PEW) [37], a condition characterized by nutritional and metabolic imbalances leading to the simultaneous depletion of body protein and energy reserves [38]. The prevalence of PEW in non-dialysis CKD populations has been reported to vary widely, ranging from18% to 75%, with increasing rates with CKD progression [39]. This broad range is attributed to the lack of standardized definitions, differences in assessment tools, small sample sizes, and the diverse socioeconomic conditions of the countries where the studies were conducted [39].

A meta-analysis examining the global prevalence of PEW using the subjective global assessment (SGA) or the malnutrition–inflammation score identified only five studies on patients with non-dialysis CKD, reporting a PEW prevalence of 11–54% [39]. Additionally, sarcopenia, a component of PEW, has now been recognized as a distinct condition with its own unique ICD code, with handgrip strength (HGS) considered as the simplest and most practical marker for its assessment. Using HGS, along with poor muscle mass and poor functional performance, the prevalence of sarcopenia in the UK Biobank CKD population was estimated at 9.7%, approximately twice the prevalence observed in individuals without CKD [40]. It is important to diagnose and treat PEW and sarcopenia in CKD, as these have been linked to adverse outcomes, including greater risk of frailty, hospitalizations, progression to kidney failure, cardiovascular adverse events, and mortality, as well as greater healthcare costs [21,40,41,42,43,44].

The etiologies of PEW and sarcopenia in CKD are multifactorial and include a hypercatabolic state induced by uremia, anorexia, and chronic inflammation [45]. CKD results in the disruption of multiple processes that normally regulate muscle metabolism [46]. CKD-related changes include mitochondrial dysfunction, resulting in inefficient energy production, inflammation, and autophagy; impaired muscle repair and regeneration due to reduced protein synthesis; and excessive muscle atrophy. Additionally, dysregulation of appetite hormones, like ghrelin and leptin, combined with inflammatory cytokines, contributes to appetite suppression and reduced food intake [46].

Metabolic acidosis, which worsens with CKD progression, further exacerbates this by promoting insulin resistance, oxidative stress, and inflammation leading to protein loss and malnutrition. Importantly, the treatment of metabolic acidosis has been shown to mitigate these effects [47,48,49].

Low-protein and very-low-protein diets, despite concerns about muscle wasting and frailty, have not been shown to accelerate their progression [46,50]. Studies suggest that patients can adapt to low-protein diets by reducing muscle protein degradation and increasing the efficiency of muscle protein turnover, ultimately preventing muscle wasting [50]. These diets slow CKD progression by reducing glomerular hyperfiltration, proteinuria, and production of uremic toxins [22,51] and by decreasing net endogenous acid production and mitigating metabolic acidosis [25] by favoring plant protein intake over animal protein.

### 4.2. Obesity Paradox in CKD

The “Obesity paradox”, where a higher body mass index (BMI) is associated with better survival outcomes, is observed in both dialysis and non-dialysis CKD populations. However, the underlying mechanisms and clinical implications remain complex and require careful interpretation.

In patients with non-dialysis CKD, several studies have demonstrated an inverse relationship between BMI and mortality. A BMI below 25 kg/m^2^ was associated with high mortality rates across all CKD stages [52]. Also, a lower BMI was associated with greater mortality in non-dialysis CKD, after adjusting for various confounders, such as demographics, comorbidities, and markers of malnutrition and inflammation [53]. Conversely, a higher BMI was associated with a reduction in all-cause mortality in CKD stages 3–5 [54,55]. The KDOQI guidelines acknowledge the complexity of the relationship between BMI and mortality in patients with non-dialysis CKD, stating that overweight or obesity status may decrease mortality risk in metabolically healthy individuals [22].

Similarly, a higher BMI in patients with kidney failure requiring hemodialysis is consistently associated with improved survival across various racial and regional groups [56]. However, the “Obesity paradox” is less consistent in patients undergoing peritoneal dialysis, where results are mixed [56,57].

Though the reasons remain uncertain, a similar “reverse epidemiology” has been noted in other chronic disease including chronic obstructive pulmonary disease, heart failure, cirrhosis and cancer [58]. Proposed explanations for the Obesity paradox in CKD include a time discrepancy in competing risk factors with high short-term mortality in advanced CKD obscuring the longer-term adverse effects of obesity; hemodynamic stability conferred by obesity; lipoprotein defense against endotoxins; protective cytokine profiles; toxin sequestration by fat mass; and antioxidation of muscle [58]. Fat mass serves as an energy reserve against catabolic conditions commonly seen in patients on hemodialysis, and higher fat mass has been associated with better survival [59]. Higher muscle mass, as indicated by serum creatinine levels, is associated with improved survival, suggesting that both fat and muscle mass contribute to the Obesity paradox in CKD [57]. The pathophysiology behind this paradox includes factors such as protein-energy wasting, inflammation, and the malnutrition–inflammation complex syndrome, which are prevalent in dialysis patients and may alter the typical risk profiles seen in the general population [60].

The potential survival benefit of mild obesity in advanced CKD must be distinguished from the well-documented renal damage caused by severe obesity, as well as the improvements in renal function observed with weight-loss interventions in individuals with mild CKD [61].

### 4.3. Nutritional Management of the Older Patient with CKD

The prevalence of CKD increases with age, reaching 33% in adults over 65 years [62]. Aging is accompanied by reductions in muscle mass and function due to increased catabolic and blunted anabolic responses in muscle metabolism, a decrease in physical activity, and reduced energy intake [63]. Additionally, older adults, particularly those with CKD, may also spontaneously reduce protein intake [64]. This increases the risk of malnutrition, PEW, and sarcopenia in the older population, which correlates with increased frailty and poorer prognosis. Thus, the KDIGO 2024 guidelines recommend individualized protein targets based on the patient’s clinical status. For older adults with stable or slowly progressing CKD, a protein intake of 1.0–1.2 g/kg/day is suggested to prevent malnutrition and sarcopenia. However, for those with significant CKD progression, protein restriction to 0.6–0.8 g/kg/day may be appropriate, provided they are metabolically stable [21,63].

Given the elevated oxidative stress in patients with CKD, clinicians should ensure adequate intake of antioxidants, such as vitamins E and C, selenium, and zinc, which are often deficient in the elderly [22], as this may help mitigate oxidative damage and support muscle health, reduce oxidative stress markers, and improve clinical outcomes [25].

Regular monitoring of nutritional status and individualizing dietary plans based on the patient’s clinical condition, preferences, and quality of life are essential. This requires a careful patient-centered, personalized approach that includes nutritional screening using validated tools, careful assessment of nutritional status, and muscle mass and function before and during dietary interventions. It also involves providing dietary counseling to ensure adequate energy and protein intakes, along with patient education and a shared decision-making process to optimize outcomes [63].

### 4.4. Sex-Related Differences in Nutrition

It is increasingly recognized that sex influences nutritional intake, nutrient metabolism, and metabolic response to dietary interventions in both humans and animals [65]. Notable differences include lower caloric consumption per kilogram of lean mass in females compared to males; higher intakes of fruits, vegetables, carbohydrates, and fiber in females; and greater consumption of fat- and protein-rich foods in males [66]. Additionally, men and women differ in their metabolic handling of glucose, lipids, and proteins, as well as in gut microbiome composition, which influence the development and treatment of conditions such as diabetes and obesity [65]. These are further influenced by changing sex hormone levels over the lifespan, such as decreased food intake during the follicular phase in women [66] and increased risks of insulin resistance and type 2 diabetes mellitus, as well as osteoporosis, after menopause [65].

Currently, CKD nutritional guidelines do not include sex-specific recommendations. However, there is increasing interest in examining sex-differences in dietary patterns and food intake [67] and the association between nutrition, muscle mass, and strength [68] in patients with CKD. Thus, current nutritional recommendations for patients with CKD do not take into account the biological differences related to nutrition in men and women, making regular monitoring of nutritional status and individualization of dietary plans, essential.

### 4.5. Fruits and Vegetables: Bad with the Good?

Metabolic acidosis, which frequently develops with worsening kidney function, promotes decline in glomerular filtration and contributes to extra-renal complications, such as muscle wasting, worsened metabolic bone disease, hypoalbuminemia, malnutrition, increased inflammation, and increased mortality [69]. To address this, KDQOI guidelines recommend decreasing net acid excretion in CKD stages 1–4 by increasing intake of fruits and vegetables as dietary sources of alkali [22]. This approach contrasts with the traditional advice to limit fruits and vegetables in advanced CKD in order to avoid hyperkalemia and hyperphosphatemia. However, studies in patients with CKD stage 3 [70] and stage 4 [71] have shown that selected patients with advanced CKD can safely incorporate fruits and vegetables under the guidance of a dietitian. Additionally, phosphorus from plant-based sources is less bioavailable than that from processed foods [22].

## 5. Implementation of Personalized Nutrition Plans

### 5.1. Collaboration with Renal Dietician and Patient Engagement

The Registered Dietitian Nutritionist (RDN) plays an important role in the kidney care team, particularly in the assessment of nutrition status and prevention of protein-energy malnutrition in dialysis populations by providing targeted interventions [72]. Early integration of RDNs in the care of patients with CKD provides an opportunity for individualized dietary planning tailored to the patient’s CKD stage and nutritional status. The KDOQI 2020 guidelines recommend medical nutrition therapy (MNT) in CKD management [22] to optimize patient outcomes, including slowing disease progression and managing comorbid conditions such as diabetes and cardiovascular disease [21]. Early engagement of RDNs ensures personalization and dynamic adjustment of dietary interventions to reflect the evolving nature of the disease, supporting better long-term outcomes [22].

The success of these interventions is significantly influenced by patient engagement. In this context, telehealth coaching provided by RDNs is a powerful tool to enhance dietary self-management in CKD populations [73]. Patients value the personalized approach, convenience, and empowerment provided by telehealth coaching, which facilitate sustainable dietary behavior changes [73]. Telehealth coaching demonstrates high retention rates with positive feedback from participants attesting to its feasibility and acceptability [74]. Alternatively, group settings, such as shared medical appointments, can foster peer support, which has been shown to improve patient outcomes in other chronic conditions [75,76,77].

### 5.2. Case Study 1: Example of a Personalized Nutrition Plan in CKD

This case highlights the impact of personalized MNT in a patient with an active lifestyle, well-controlled type 2 diabetes and hypertension, CKD stage 3B, and a recent diagnosis of renal cell carcinoma (Figure 2). The patient was referred for nephrology care for the management of CKD 3B. A standard CKD diet was recommended, limiting protein to 0.8 g/kg/day (60 g/day) and 2000 kcal/day. In the setting of ongoing cancer, regular vigorous exercise, and dietary restrictions, the patient lost 10 pounds of weight over a couple of weeks. Recognizing the nutritional risks posed by cancer and unintentional weight loss, the nephrologist referred the patient to a dietitian specializing in CKD for a tailored intervention.

The initial nutrition assessment was conducted eight days after the referral via telehealth, for patient convenience. The patient and his wife shared their concerns about balancing dietary restrictions with maintaining nutrition. The dietitian implemented a personalized plan focusing on weight stabilization, recommending increased caloric intake (2300–3000 kcal/day) and sodium restriction (2000 mg/day). Protein intake was adjusted to a minimum of 0.9 g/kg/day (70 g/day) to address ongoing weight loss, emphasizing plant-based proteins and healthy fats while prioritizing caloric adequacy over strict adherence to renal diet parameters.

Following the nephrectomy for renal cell carcinoma, the patient resumed physical activity (playing tennis for one hour, two or three days a week), and continued close follow-up with the dietitian. As the cancer treatment progressed, the dietitian advised further liberalization of protein intake (~80 g/day) and inclusion of whole grains to support caloric needs and glycemic control, coordinated with the patient’s nephrologist and oncologist. Over the next year, the patient achieved stable nutritional status, gaining back 10 pounds and maintaining normal serum phosphorus, potassium, and albumin levels.

This case underscores the critical role of early, personalized MNT in managing complex concurrent medical conditions like cancer. The tailored approach allowed for flexibility in dietary recommendations to mitigate nutritional risks associated with cancer treatment, while continuous communication with the dietitian provided reassurance and facilitated adherence to an evolving dietary plan.

### 5.3. Case Study 2: Example of a Personalized Nutrition Plan for an Older Female Patient with Advanced CKD, Food Intolerances, and Unintentional Weight Loss

This case highlights the importance of personalized MNT in limiting unintentional weight loss and protein calorie malnutrition in a postmenopausal woman. An 83-year-old woman with stage 4 CKD, osteoporosis, and irritable bowel syndrome (IBS) with chronic diarrhea was referred to the dietician after experiencing an unintentional loss of 32% of her body weight over the preceding four years. To alleviate the diarrhea, the patient was interested in trying a low-FODMAP (fermentable oligosaccharides, disaccharides, monosaccharides, and polyols) diet. She had been focusing on limiting salt intake due to her kidney disease and was concerned that additional restrictions that needed to follow the low-FODMAP diet would eliminate nutrient-rich foods and limit dairy consumption, which would impact her osteoporosis. A dietary recall revealed that the patient was intolerant to high-fat foods and gluten-free fructans. She was advised to limit fat intake to 40–50 g per day and reduce foods rich in gluten-free fructans. To maintain adequate calorie and calcium intakes, she was encouraged to consume lower-fat dairy products. Additionally, she was recommended to increase the consumption of starchy foods like boiled potatoes, white rice, and white bread as primary caloric sources and to continue to limit sodium intake to 2000 mg per day.

She was also encouraged to maintain a consistent eating schedule with four meals per day, including a light breakfast, dinner, and a snack, in addition to lunch, which was her main meal of the day. This approach aimed to facilitate digestion and achieve adequate caloric intake. With these interventions, her weight and kidney function remained stable, while her potassium, phosphorus, and albumin levels stayed within normal ranges.

This case highlights the role of personalized MNT in stopping ongoing involuntary weight loss and preventing malnutrition while managing the complications of advanced CKD. It also demonstrates how a tailored dietary approach can help balance nutritional needs dictated by multiple comorbidities while allowing the patient to enjoy some of their favorite food.

### 5.4. Barriers to Nutrition Plans Implementation

There are several barriers to referral to RDNs for nutrition assessment and interventions for CKD [78]. One such barrier is the knowledge of non-dietetic health professionals regarding the importance of diet in managing CKD. A study found that non-dietetic health professionals rated the importance of diet significantly lower than renal dietitians [78], and about 50% of them referred patients to renal dietetic services infrequently, citing reasons such as a perceived lack of evidence supporting the role of dietary interventions in slowing CKD progression, concerns about patient adherence, and a desire to reduce visit burden for patients [78]. Another barrier is the inadequate staffing of dietetic services, leading to lengthy wait times for patients. The model of shared medical appointment can help address this issue and reduce wait times [79]. Additionally, the availability of dietitians, particularly in low- and middle-income countries, is limited and dietary counseling is often unavailable [80]. Despite strong interest in medical nutrition therapy among patients with CKD, practical challenges related to cost, insurance coverage, and reimbursement continue to pose significant barriers to its accessibility [81].

## 6. Individual Variability in Dietary Response: The Role of Epigenetics and Precision Nutrition in CKD

The response to dietary interventions varies significantly among individuals [82]. This variability is driven by complex interactions between genetic, epigenetic, metabolic, and environmental factors [83,84]. Nutrigenetics examines how genetic variations influence an individual’s response to dietary intake and susceptibility to nutrition-related diseases, while nutrigenomics focuses on how nutrients affect gene expression, impacting the proteome and metabolome [85]. Together, these disciplines provide a foundation for precision nutrition in CKD care.

Epigenetic modifications, including DNA methylation, histone modifications, and non-coding RNA activity, are critical regulators of gene expression and contribute to the heterogeneity of dietary responses in CKD [84,86,87,88]. For example, histone acetylation promotes gene expression, while deacetylation represses it [84]. Nutrients such as butyrate, a histone deacetylase inhibitor, have shown potential in modulating these modifications to ameliorate CKD progression [88]. Altered DNA methylation patterns in genes, such as interleukin 6, transforming growth factor beta1, and paraoxonase 1, which are associated with inflammation, oxidative stress, and fibrosis, can influence an individual’s response to dietary sodium and protein restrictions [86,88].

### 6.1. Metabolomics and Proteomics

Metabolomic and proteomic profiling may also help clarify patient-specific dietary responses. Biomarkers such as trimethylamine-N-oxide (TMAO) and urea cycle intermediates have been identified as potential indicators of how patients respond to protein intake adjustments. Tailored dietary interventions, such as reducing intake of choline- and carnitine-rich foods (e.g., red meat and eggs), can be implemented to lower TMAO levels and mitigate cardiovascular risk [89,90]. Additionally, urea cycle intermediates, such as citrulline and arginine, are also important biomarkers in CKD. Dietary protein restriction can help manage urea cycle dysregulation, but the response to such interventions can vary [91].

### 6.2. Lipidomics

CKD leads to alterations in the concentration and structure of numerous lipid classes [92], including dysfunctional high-density lipoproteins (HDLs) enriched with ceramides and sphingomyelins, which contribute to a pro-inflammatory and pro-atherogenic state [93]. Impaired fatty acids beta-oxidation leads to the accumulation of saturated free fatty acids and long-chain acylcarnitines [94], shifting the balance from decreased levels of poly unsaturated fatty acids (PUFAs) in early-stage CKD toward increased short-chain and saturated fatty acids in late-stage CKD [92,94,95]. Additionally, lipids in circulation can accumulate in the kidneys resulting in lipotoxicity, leading to oxidative stress, inflammation, fibrosis and albuminuria [96]. Though there has been extensive research regarding the role of lipidomics biomarkers in kidney disease [97], evidence that directly translates lipidomic analyses to tailored nutritional interventions in CKD is lacking. Nonetheless, lipidomics is a promising and evolving field, and future investigations should clarify how lipid profile alterations can guide dietary recommendations in CKD.

### 6.3. Genetic Integration

The integration of genetic data into dietary planning has revealed several candidate genes influencing CKD progression and response to diet. Variants in UMOD (Uromodulin) and SHROOM3 (SHROOM family member 3) genes have been associated with CKD progression and response to dietary intervention, suggesting their potential as markers for guiding sodium and protein intakes [98,99]. Similarly, APOL1 variants, prevalent in individuals of African descent, affect dietary sodium sensitivity and protein metabolism, emphasizing the importance of population-specific considerations in precision nutrition [100]. Polymorphism in MTHFR (methylenetetrahydrofolate reductase) gene impacts homocysteine metabolism, which is relevant for patients with CKD because of its role in cardiovascular risk and response to folate intake [98]. Variants in SOD2 (superoxide dismutase 2), an antioxidant enzyme, are associated with oxidative stress and inflammation in CKD and can guide antioxidant supplementation [101]. Lastly, the CYP11B2 (cytochrome P450 family 11 subfamily B member 2) gene is part of the renin–angiotensin–aldosterone system, and its variants are associated with hypertension and CKD. These markers can guide dietary sodium and potassium management [98].

Precision nutrition strategies aim to leverage these insights by developing individualized dietary plans that account for genetic predispositions, epigenetic modifications, and biomarker profiles. For example, a patient with CKD and a high-risk APOL1 genotype may benefit from stricter sodium restrictions and protein modifications to mitigate disease progression. Similarly, incorporating metabolomic data can guide adjustments in micronutrient intake to address deficiencies or toxicities unique to the individual.

However, the high costs and complexity of genetic and metabolomic testing limit its accessibility in routine clinical practice. Furthermore, long-term data on the effectiveness of precision nutrition for CKD are scarce, with most studies focusing on short-term outcomes such as biomarker changes rather than clinically meaningful endpoints like CKD progression or cardiovascular events.

Despite the potential benefits of genetic testing in CKD management, its clinical implementation remains limited. Many nephrologists underutilize genetic testing due to perceived barriers, including cost, complexity, and lack of familiarity with genetic data interpretation [102]. While broad genetic panels, such as the 385-gene panel from the Renasight study, have shown utility in improving diagnostic precision and guiding management, their integration into routine practice is still in its early stages and requires further validation and standardization [103].

Addressing these gaps requires robust, multi-center longitudinal trials that integrate genetic, epigenetic, and metabolomic data with real-world patient outcomes. The integration of omics technologies with cultural and socioeconomic considerations is essential to ensure that dietary interventions are both effective and equitable.

## 7. Conclusions

The application of precision nutrition in CKD represents a transformative approach to dietary management, addressing the heterogeneity of patient needs. By tailoring interventions to the individual, this strategy has the potential to optimize outcomes, mitigate risks, and improve the quality of life for patients with CKD. Sustained research efforts and advancements in implementation science are needed to fully address the potential of personalized nutrition in this complex and vulnerable population.

## Figures and Tables

**Figure 1 biomedicines-13-00647-f001:**
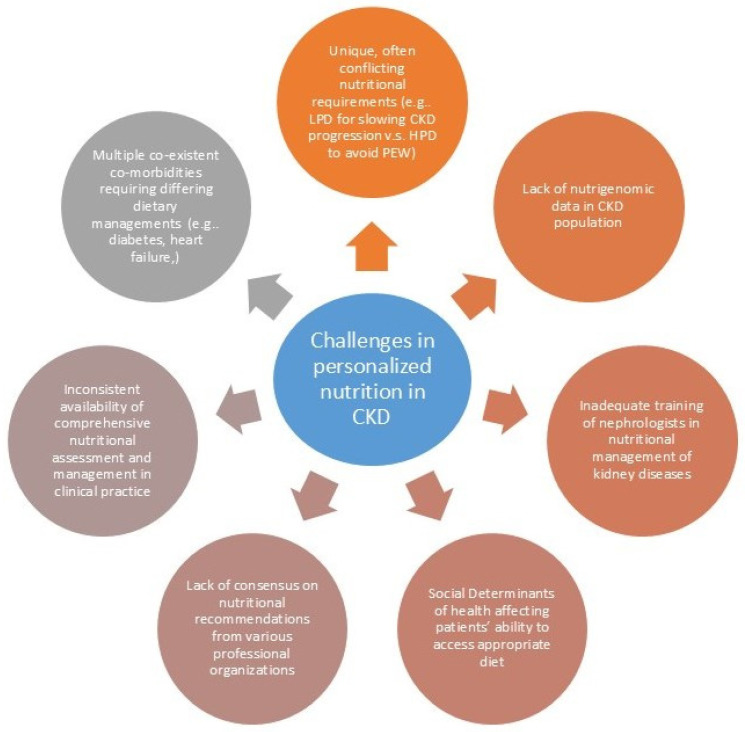
Challenges to optimal nutrition in CKD. CKD = chronic kidney disease; LPD = low-protein diet; HPD = high-protein diet; PEW—protein-energy wasting.

**Figure 2 biomedicines-13-00647-f002:**
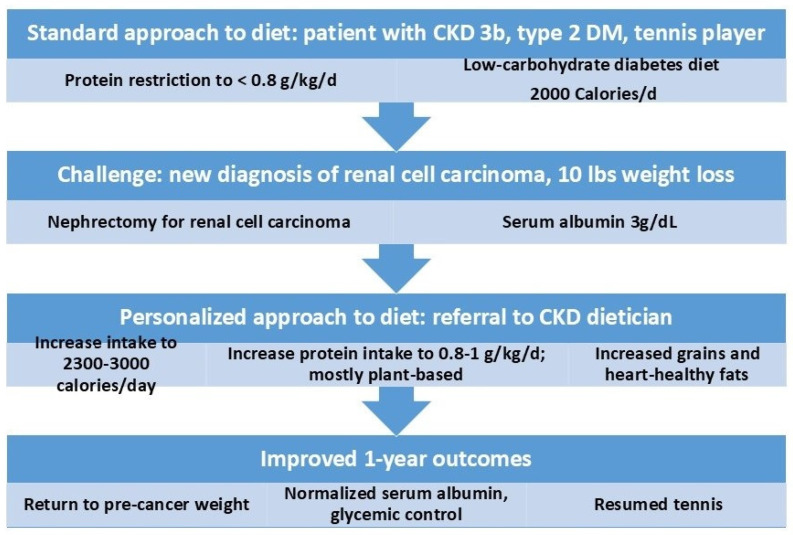
Case study showing importance of medical nutrition therapy in managing complex co-morbidities. CKD = chronic kidney disease; DM = diabetes mellitus.

**Table 1 biomedicines-13-00647-t001:** Overview of the types of diets in chronic kidney disease.

Type of Diet	Advantages	Disadvantages
Plant-based diet	Helps control metabolic acidosis, proteinuria and inflammation; supports the microbiome and potentially slows CKD progression. Associated with lower risks of hypertension, diabetes mellitus, cardiovascular disease and mortality.	Careful monitoring of potassium and phosphorus levels is required in advanced stages of CKD. Deficiencies of essential amino acids, vitamin B12 and long chain n-3 polyunsaturated fatty acids possible.
Mediterranean diet	Cardioprotective and anti-inflammatory effects. It may also improve kidney function and metabolic profile and reduce inflammation and uremic toxin production.	May require potassium monitoring in advanced CKD.
Low-protein or controlled-protein diet	Limited protein intake is intended to reduce nitrogenous waste generation and slow kidney function decline. Non-dialysis CKD: 0.6–0.8 g/kg/day Kidney disease requiring dialysis: 1.0–1.2 g/kg/day	Excessive protein restriction can lead to malnutrition and muscle wasting, specially in patients with advanced CKD.
Very low-protein diet (0.3 g/kg/day) with ketoanalogue supplementation	Can be supplemented with essential amino acids and ketoacids to potentially delay the progression of CKD.	Requires careful monitoring and supplementation to avoid deficiencies and ensure nutritional adequacy.
Low-sodium diet	Controls blood pressure and reduces cardiovascular risk.	Challenging to adhere to.
Intermittent fasting	May have benefits for metabolic health, though limited research exists on it’s risks or benefits for patients with CKD.	Risks include nutrient deficiencies, dehydration and difficulty in adherence, specially in patients with advanced CKD.

## Data Availability

No new data were created or analyzed in this study.

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
