# Peer review of "Personalized Nutrition in Chronic Kidney Disease"

_biomedicines, 2025, doi:10.3390/biomedicines13030647_

Round 1

Reviewer 1 Report

Comments and Suggestions for Authors

The paper Is interesting. The paper gives a large overview in the interesting field of nutrition in ckd patients. The topic Is centrale for the treatment of CKD patients whose worsening Is often due to malnutrition. A personalized approach to the patients Is the most correct and severe limitations of caloric and proteic intake are very Dangerous.

On my opinion, the description of the case Is very brief, although interesting. Authors add two cases and this should be enlarged.
Figures are descriptive. For me, Is an interesting review worthy of publication After minor revision.

Author Response

Comment 1: The authors must explain the exclusion of lipids and lipidomics from the personalised nutrition approach to CKD. This is a decision that makes the review very weak, considering the recent advances of the inflammatory status and lipidomic analyses for this disease condition. Response: We agree with the Editor that lipidomic analyses have shown promise in elucidating aspects of CKD pathophysiology, with a particular focus on biomarker discovery and mechanistic studies. Our decision to exclude lipidomics from this review was driven by the lack of evidence that directly translates lipidomic findings into personalized dietary interventions for patients with CKD. Because our aim was to highlight established and emerging ‘omics’-driven strategies for customized nutritional guidance, we opted to focus on areas where evidence directly supports practical, patient-centered dietary modifications. We agree that lipidomics is a promising and evolving field and acknowledge that future research will likely delineate how lipid profile alterations can guide dietary choices in CKD. We added the following paragraph on page 12:  

Lipidomics:“CKD leads to alteration in the concentration and structure of numerous lipid classes, including dysfunctional high-density lipoproteins (HDL) enriched with ceramides and sphingomyelins which contributes to a pro-inflammatory and pro-atherogenic state. Impaired fatty acids beta-oxidation leads to the accumulation of saturated free fatty acids and long-chain acylcarnitines, shifting the balance from decreased level of poly unsaturated fatty acids (PUFA) in early-stage CKD towards increased in short-chain and saturated fatty acids in late-stage CKD. Though there has been extensive research regarding the role of lipidomics biomarkers in kidney disease, evidence that directly translates lipidomic analyses to tailored nutritional interventions in CKD is lacking. Nonetheless, lipidomics is a promising and evolving field, and future investigations should clarify how lipid profile alterations can guide dietary recommendations in CKD.” 

Reviewer 2 Report

Comments and Suggestions for Authors

The abstract should be rewritten to include information about the aim of the work, results and conclusions?

What was the aim of the review article? Did the authors plan a systematic review? Too few references to other recommendations for CKD.

If the authors propose new solutions, they should base them on a systematic review or even design such a study or case study.

Table 1 - no source

Figure 2 - illegible, no source

Figure 3 - no source

An interesting work requires substantive development. The chapters only indicate issues but do not exhaust them.

No conclusions and limitations of the study.

Author Response

The abstract should be rewritten to include information about the aim of the work, results and conclusions?

Response:

We have revised the abstract to better reflect the scope of the review, as follows:

 A personalized approach to nutrition in patients with chronic kidney disease (CKD) represents a promising paradigm shift in disease management, moving beyond traditional one-size-fits-all dietary recommendations. CKD patients often have other comorbidities and face unique nutritional challenges, including protein-energy wasting (PEW), sarcopenia, and impaired renal excretion of nutrients, which complicate dietary planning. Current guidelines focus primarily on nutrient restrictions—such as limiting protein, sodium, potassium, and phosphorus. However, these generalized recommendations often result in suboptimal adherence and outcomes. Personalized nutrition, which adapts dietary recommendations to individual characteristics such as genotype, phenotype, and socio-cultural preferences, has gained traction across various chronic diseases. However, its application in nephrology remains underexplored, and despite promising results from studies such as Food4Me, questions remain about the real-world impact of such strategies.

The aims of this review are 1) to summarize the evidence on the current state of nutritional recommendations in CKD, 2) to discuss the emerging role of multi-omics approaches in informing personalized nutrition advice in CKD, and 3) to provide an opinion on nutritional challenges faced by CKD patients and the importance of collaboration with the renal dietician.

We conclude that despite barriers such as the cost and data integration, personalized nutrition holds the potential to improve CKD outcomes, enhance quality of life, and empower patients through tailored dietary strategies for better disease management.

What was the aim of the review article? Did the authors plan a systematic review? Too few references to other recommendations for CKD.

Response:

In our review titled “Personalized approach to nutrition in CKD”, we are solely focusing on nutritional guidelines and a personalized approach to nutrition in CKD. Is not intended to be a systematic review, nor is intended to review other recommendations for CKD treatment. We have clarified the scope in the abstract, and at the end of the introduction paragraph as below:

Abstract:

A personalized approach to nutrition in patients with chronic kidney disease (CKD) represents a promising paradigm shift in disease management, moving beyond traditional one-size-fits-all dietary recommendations. CKD patients often have other comorbidities and face unique nutritional challenges, including protein-energy wasting (PEW), sarcopenia, and impaired renal excretion of nutrients, which complicate dietary planning. Current guidelines focus primarily on nutrient restrictions—such as limiting protein, sodium, potassium, and phosphorus. However, these generalized recommendations often result in suboptimal adherence and outcomes. Personalized nutrition, which adapts dietary recommendations to individual characteristics such as genotype, phenotype, and socio-cultural preferences, has gained traction across various chronic diseases. However, its application in nephrology remains underexplored, and despite promising results from studies such as Food4Me, questions remain about the real-world impact of such strategies.

The aims of this review are 1) to summarize the evidence on the current state of nutritional recommendations in CKD, 2) to discuss the emerging role of multi-omics approaches in informing personalized nutrition advice in CKD, and 3) to provide an opinion on nutritional challenges faced by CKD patients and the importance of collaboration with the renal dietician.

We conclude that despite barriers such as the cost and data integration, personalized nutrition holds the potential to improve CKD outcomes, enhance quality of life, and empower patients through tailored dietary strategies for better disease management.

…and at the end of Introduction, on page 2:

“…This review assesses the current state of nutritional guidelines for the management of CKD, the emerging role of multi-omics in informing personalized nutrition advice in CKD, and highlights the associated nutritional challenges. It further explores the barriers to implementing personalized nutrition strategies in clinical practice for CKD patients and outlines a potential framework for advancing individualized nutritional approaches to optimize CKD management on a patient-specific basis via two vignettes.”

If the authors propose new solutions, they should base them on a systematic review or even design such a study or case study.

Response:

 Proposing new solutions or designing a new study based on a systematic review was outside of the scope of this manuscript.

Table 1 - no source

Figure 2 - illegible, no source

Figure 3 - no source

Response:

Table 1 and Figures 2 and 3 have been created by us based on all the literature discussed in the review, as well as on our opinion and clinical expertise. No source other than the authors of the manuscript exists.

An interesting work requires substantive development. The chapters only indicate issues but do not exhaust them. No conclusions and limitations of the study.

Response:

The conclusions are provided at the end of the abstract and in the last paragraph of the manuscript. As this review focuses on summarizing current evidence and offering our perspective on advancing personalized nutritional approaches in CKD, a limitations section is not applicable, as we are not presenting original data or novel methodologies.

Reviewer 3 Report

Comments and Suggestions for Authors

The authors prepared a well-written review about personalized nutrition in chronic kidney disease patients. In this review, they systematically presented current knowledge about this scientifically important topic. The authors have summarized the types of diets that are recommended in this moment and the challenges that have to be overcome to achieve optimal nutrition in chronic kidney patients.

1. As I minor revision suggestion, I would recommend to the authors to carefully check typing as there are some mistakes that are repeating through all the text such as . or , followed by number of references (for example, line 37...syndrome. [1] should be replaced by syndrome [1].; line 43 ... medicine. [4,5] should be replaced by medicine [4, 5]. line 48 metabolic syndrome, [8] should be replaced by metabolic syndrome [8], ...).

2. I suggest to authors to change the title of Table 1 maybe to Overview of...

3. Under the Figure 1. the abbreviation for PEW is missing.

4. lines 246, 254 "Obesity paradox" should stand instead of obesity paradox

5. lines 385 Descriptions for abbreviations TGFB1 and PON1 are missing.

Kind regards

Author Response

The authors prepared a well-written review about personalized nutrition in chronic kidney disease patients. In this review, they systematically presented current knowledge about this scientifically important topic. The authors have summarized the types of diets that are recommended in this moment and the challenges that have to be overcome to achieve optimal nutrition in chronic kidney patients.

  1. As I minor revision suggestion, I would recommend to the authors to carefully check typing as there are some mistakes that are repeating through all the text such as . or , followed by number of references (for example, line 37...syndrome. [1] should be replaced by syndrome [1].; line 43 ... medicine. [4,5] should be replaced by medicine [4, 5]. line 48 metabolic syndrome, [8] should be replaced by metabolic syndrome [8], ...).
  2. I suggest to authors to change the title of Table 1 maybe to Overview of...
  3. Under the Figure 1. the abbreviation for PEW is missing.
  4. lines 246, 254 "Obesity paradox" should stand instead of obesity paradox
  5. lines 385 Descriptions for abbreviations TGFB1 and PON1 are missing.

Kind regards

Response:

Thank you for your kind comments and valuable feedback. We have corrected the typographical errors, modified the title of Table 1, added the expansions for all of the noted abbreviations and changed the capitalization as recommended.

Reviewer 4 Report

Comments and Suggestions for Authors

The article Personalized Nutrition in Chronic Kidney Disease is very well written and structured. There is a clear and significant need for precision nutrition, particularly considering individual variability. I also appreciated the case study included.

That said, it may be beneficial to include additional examples, particularly given the anatomical and physiological differences between men and women. For instance, the proportion of muscle mass can differ significantly. Additionally, I would be interested in understanding how the standard approach to diet can be adjusted for conditions such as menopause and andropause. What role does physical activity play for these patients—should it be encouraged, or is it contraindicated? Furthermore, I am curious about the strategies currently applied to address sarcopenia in these individuals.

Author Response

The article Personalized Nutrition in Chronic Kidney Disease is very well written and structured. There is a clear and significant need for precision nutrition, particularly considering individual variability. I also appreciated the case study included.

That said, it may be beneficial to include additional examples, particularly given the anatomical and physiological differences between men and women. For instance, the proportion of muscle mass can differ significantly. Additionally, I would be interested in understanding how the standard approach to diet can be adjusted for conditions such as menopause and andropause.

Response:

We thank the reviewer for the positive comments and suggestions for improvement. We agree that differences in physiology and metabolism between the two sexes and the effect of hormonal changes overtime are important issues to examine. We have added the following paragraph discussing sex related differences in nutrition on page 8.

Sex related differences in nutrition

It is increasingly recognized that sex influences nutritional intake, nutrient metabolism and metabolic response to dietary interventions in both humans and animals [61]. Notable differences include lower calorie consumption per kilogram of lean mass in females compared to males, higher intake of fruits, vegetables, carbohydrates and fiber, in females, and greater consumption of fat and protein rich foods in males [62]. Additionally, men and women differ in their metabolic handling of glucose, lipids and proteins, as well as in gut microbiome composition which influence development and treatment of conditions such as diabetes and obesity[61]. These are further influenced by changing sex hormone levels over the lifespan, such as decreased food intake during follicular phase in women [Marino] and increased risk of insulin resistance and type 2 diabetes mellitus and osteoporosis after menopause [61].

Currently, CKD nutritional guidelines do not include sex-specific recommendations. However, there is increasing interest in examining sex-differences in dietary patterns and food intake [63] and the association between nutrition, muscle mass and strength [64] in CKD patients. Thus current nutritional recommendations for patients with CKD do not take into account the biological differences related to nutrition in men and women, making  regular monitoring of nutritional status and individualization of dietary plans, essential.

What role does physical activity play for these patients—should it be encouraged, or is it contraindicated? Furthermore, I am curious about the strategies currently applied to address sarcopenia in these individuals.

The first case we present reflects adjustments made to diet in the form of increasing the caloric intake in an active man who performs vigorous physical activity (tennis) on a regular basis. However, a comprehensive review of physical activity recommendations in CKD is beyond the scope of our current work. For the same reason, the strategies currently applied to address sarcopenia in CKD individuals are also outside the scope of this review. We did however include a second clinical vignette to reflect challenges to nutritional approach in a post-menopausal  woman with advanced CKD and severe malnutrition on page 11 as below:

Case study 2: Example of personalized nutrition plan in an older female patient with advanced CKD, food intolerances and unintentional weight loss

This case highlights the importance of personalized MNT in limiting unintentional weight loss and protein calorie malnutrition in a postmenopausal woman. An 83 year old woman with CKD stage 4, osteoporosis and irritable bowel syndrome (IBS) with chronic diarrhea was referred to the dietician after experiencing unintentional loss of 32% of her body weight over preceding four years. To alleviate the diarrhea, the patient was interested in trying a low-FODMAP (Fermentable Oligosaccharides, Disaccharides, Monosaccarides and polyols) diet. She had been focusing on limiting salt intake due to her kidney disease and was concerned that additional restrictions needed to follow the low-FODMAP diet would eliminate nutrient rich foods and limit dairy consumption, which would impact her osteoporosis. A dietary recall revealed that the patient was intolerant to high-fat foods and in gluten-free fructans. She was advised to limit fat intake to 40-50 grams per day and reduce foods rich in gluten-free fructans. To maintain adequate calorie and calcium intake, she was encouraged to consume lower-fat dairy products. Additionally, she was recommended to increase the consumption of starchy foods like boiled potatoes, white rice and white bread as a primary caloric source and to continue to limit sodium intake to 2000 mg per day.

She was also encouraged to maintain a consistent eating schedule with four meals per day, including a light breakfast, dinner and a snack, in addition to lunch, which was her main meal of the day. This approach aimed to facilitate digestion and achieve adequate caloric intake. With these interventions, her weight and kidney function remained stable, while her potassium, phosphorus and albumin levels stayed within normal ranges.

This case highlights the role of personalized MNT in stopping ongoing involuntary weight loss and preventing malnutrition while managing the complications of advanced CKD. It also demonstrates how a tailored dietary approach can help balance nutritional needs dictated by multiple comorbidities, while allowing the patient to enjoy some of their favorite food.

Round 2

Reviewer 2 Report

Comments and Suggestions for Authors

In my opinion, the authors made necessary corrections according to my remarks. Thank you.

Author Response

Comments and Suggestions for Authors

In my opinion, the authors made necessary corrections according to my remarks. Thank you.

Response: Thank you.